**Data Availability Statement:** All datasets supporting the conclusions of the present study are obtained from the MIMIC-IV database (website: https://physionet.org/content/mimiciv/).

# Association of red blood cell distribution width-to-albumin ratio with mortality in patients undergoing transcatheter aortic valve replacement

**Limin Meng**[1,2☯], **Hua Yang**[1,2☯], **Shuanli Xin**[2], **Chao Chang**[2], **Lijun Liu**[2], **Guoqiang Gu**[1]*

1 Department of Cardiology, The Second Hospital of Hebei Medical University, Shijiazhuang, Hebei, China,
2 Department of Cardiology, Handan First Hospital, Handan, Hebei, China

☯ These authors contributed equally to this work.
* guguoqiang2022@163.com

## Abstract

### Background

Frailty is associated with poor prognosis in patients undergoing transcatheter aortic valve replacement (TAVR). The red blood cell distribution width (RDW)-to-albumin ratio (RAR) reflects key components of frailty. This study aimed to evaluate the relationship between RAR and all-cause mortality in patients undergoing TAVR.

### Methods

The data were extracted from the Medical Information Mart for Intensive Care IV database. The RAR was computed by dividing the RDW by the albumin. The primary outcome was all-cause mortality within 1-year following TAVR. The association between RAR and the primary outcome was evaluated using the Kaplan-Meier survival curves, restricted cubic spline (RCS), and Cox proportional hazard regression models.

### Results

A total of 760 patients (52.9% male) with a median age of 84.0 years were assessed. The Kaplan-Meier survival curves showed that patients with higher RAR had higher mortality (log-rank $P < 0.001$). After adjustment for potential confounders, we found that a 1 unit increase in RAR was associated with a 46% increase in 1-year mortality (HR = 1.46, 95% CI:1.22–1.75, $P < 0.001$). According to the RAR tertiles, high RAR (RAR > 4.0) compared with the low RAR group (RAR < 3.5) significantly increased the risk of 1-year mortality (HR = 2.21, 95% CI: 1.23–3.95, $P = 0.008$). The RCS regression model revealed a continuous linear relationship between RAR and all-cause mortality. No significant interaction was observed in the subgroup analysis.

**Funding:** The authors received no specific funding for this work.

**Competing interests:** The authors have declared that no competing interests exist.

## Conclusion

The RAR is independently associated with all-cause mortality in patients treated with TAVR. The higher the RAR, the higher the mortality. This simple indicator may be helpful for risk stratification of TAVR patients.

## Introduction

With the improvement of operator techniques and medical technology, the indications for transcatheter aortic valve replacement (TAVR) are expanding to patients with moderate and low surgical risk, making it an acceptable alternative to surgical aortic valve replacement [1–3]. Although TAVR benefits many patients in terms of survival and symptoms, some patients have died or been readmitted within one year of the procedure [4, 5]. Therefore, as more patients qualify for TAVR, identifying high-risk patients becomes increasingly essential. This can assist clinicians in risk-stratifying TAVR patients and making timely individualized interventions.

Frailty is an aging syndrome that frequently occurs in elderly individuals and increases the risk of mortality and disability in the population undergoing TAVR [6, 7]. Accurate frailty assessment may reduce mortality and readmissions after TAVR. Studies have shown that inflammation is linked with the progression of aortic valve disease and frailty [8–10]. Recent research has found a link between red cell distribution width (RDW) and the body's systemic inflammatory response [11]. Moreover, RDW was an independent predictor of frailty in older adults with coronary heart disease [12]. Aung N et al. also reported that a baseline of higher RDW and growing RDW were substantially associated with a higher risk of mortality in TAVR patients [13]. Besides, albumin, a known marker for frailty, was related to malnutrition and chronic inflammation [14]. Previous research has found that hypoalbuminemia is a powerful predictor of post-TAVR death [15].

The RDW-to-albumin ratio (RAR), an innovative and comprehensive biomarker, combines systemic inflammation and nutritional status to reflect essential aspects of frailty. Linking RDW and nutritional parameters better reflects the geriatric characteristics of these patients. RAR is a better prognostic marker than either albumin or RDW alone in patients with heart failure, stroke, cancer, aortic aneurysms, diabetic ketoacidosis, and diabetic foot [16–22]. However, the relationship between RAR and outcomes in patients undergoing TAVR is uncertain. Therefore, this study aimed to explore the association between RAR and all-cause mortality in patients with TAVR and to evaluate whether RAR contributes to risk stratification.

## Materials and methods

### Data source

We used the Medical Information Mart for Intensive Care IV (MIMIC-IV version 2.0) database, a publicly available database of health-related information on patients hospitalized at the Beth Israel Deaconess Medical Center between 2008 and 2019 [23]. We finished the Protecting Human Research Participants training course on the National Institutes of Health website to gain access to the database (certificate number: 46962361). The study was approved and granted a waiver of informed consent by the institutional review boards of the Massachusetts Institute of Technology (MIT) and Beth Israel Deaconess Medical Center (BIDMC). The study

followed the Declaration of Helsinki guidelines and the Strengthening the Reporting of Observational Studies in Epidemiology (STROBE) reporting guidelines.

## Study population

We used the International Classification of Diseases, Tenth Revision, Clinical Modification (ICD-10-CM) procedural codes (02RF37Z, 02RF38Z, 02RF3JZ, 02RF3KZ, or X2RF332) and ICD-9-CM procedural codes of 35.05 to identify patients who underwent TAVR. If patients underwent repeat TAVR, only patients with the first procedure were included. Patients with missing RDW or albumin data were excluded.

## Study variables

Based on clinical experience or previous research, the following baseline characteristics were taken, including demographic information, routine blood, biochemistry, laboratory indicators affecting RDW or albumin, and comorbidities. Blood tests performed within 48h before TAVR were used. In addition, variables that had missing values of more than 10% were excluded. The RAR was computed by dividing the RDW (%) by the albumin (g/dL). Comorbidities identified based on documented ICD-9 or ICD-10 diagnostic codes included hypertension, myocardial infarction, congestive heart failure, atrial fibrillation/flutter, diabetes mellitus, renal disease, peripheral vascular disease, pulmonary disorder, rheumatic disease, liver disease, and cerebrovascular disease.

## Primary outcome and secondary outcome

The primary outcome was all-cause mortality within 1-year following TAVR. The secondary outcome was the length of hospital stay. Data on mortality was gathered from the US Social Security Death Index for discharged patients. No patient was lost to follow-up.

## Statistical analysis

The included patients were divided into three groups according to the tertiles of RAR level. Continuous variables were presented as means ± standard deviations (SD) or median and interquartile ranges (IQR) and categorical variables as frequencies and percentages (%). Baseline characteristics were compared using the one-way ANOVA tests or Kruskal-Wallis H-tests for continuous variables and the chi-square or Fisher's exact test for categorical variables. For variables with missing values under 5%, the corresponding mean or median was imputed.

We estimated the association between RAR and the primary outcome using the univariable and multivariable Cox proportional hazards regression models. The results were presented as hazard ratios (HR) with 95% confidence intervals (CI). The following principle was used to determine whether to adjust the covariates: when added to the model, the matched HR would change by at least 10%. Model 1: no covariate adjustment; Model 2: age, gender, hemoglobin, mean corpuscular hemoglobin concentration (MCHC), blood urea nitrogen, chloride, congestive heart failure, hypertension, atrial flutter/fibrillation, diabetes with complications, and renal disease adjustment. The variance inflation factor (VIF) determined the collinearity of parameters. The proportional hazards assumption was validated using the cox.zph function from the R(survival) package. Moreover, the association between RAR and the primary outcome was outlined using the restricted cubic spline (RCS) regression model. We also performed subgroup analyses based on age ($\leq 84$ and $> 84$ years, median), gender, anemia (hemoglobin men $<13$ g/dL, women $<12$ g/dL), congestive heart failure, renal disease, diabetes, hypertension, and atrial flutter/fibrillation to identify the stability. The interaction between

RAR and stratified variables was also investigated further. In addition, we performed a sensitivity analysis, excluding deaths occurring within 30 days of enrolment to exclude deaths that could potentially be attributable to a technical complication or an acute deterioration. The relationship between RAR and length of hospital stay was analyzed using the multivariable linear regression model, with the results expressed as β (95% CI).

All analyses were performed using R software version 3.6.3. A two-sided *P* value of 0.05 was considered statistically significant.

## Results

### Patients characteristics

There were 760 TAVR patients in our research. Based on the RAR value, the participants in this study were divided into low ($< 3.5$), middle (3.5–4.0), and high ($> 4.0$) RAR groups. Table 1 shows the baseline characteristics of the groups. Their median age was 84.0 (77.0, 88.0) years old, among whom about 402 (52.9%) were male.

In summary, the higher RAR group had higher RDW, platelet, creatinine, anion gap, blood urea nitrogen, and Charlson comorbidity index but lower levels of albumin, RBC, MCHC, hematocrit, hemoglobin, mean corpuscular volume, mean corpuscular hemoglobin, chloride, and sodium. Furthermore, patients with higher RAR were more prone to having congestive heart failure, atrial fibrillation/flutter, pulmonary disorder, rheumatic disease, liver disease, diabetes with complications, and renal disease than those with lower RAR. Notably, gender, age, and BMI did not significantly differ among the groups. Patients with the higher RAR has higher all-cause mortality, and longer length of hospital stay (all $P < 0.001$).

### Association between RAR and mortality

Table 2 displays the results of univariable and multivariable Cox regression analyses. In the univariable analysis, RAR was substantially associated with all-cause mortality (HR = 1.78, 95% CI: 1.53–2.07, $P < 0.001$). The association did not change significantly after adjusting for potential confounders in Model 2, and the RAR remained an independent predictor of mortality (HR = 1.46, 95% CI:1.22–1.75, $P < 0.001$). After excluding patients who died within 30 days, the adjusted association remained significant (HR = 1.39, 95% CI: 1.08–1.79, $P = 0.010$) (S1 Table).

We also converted RAR from a continuous variable to a categorical variable (tertile) to conduct the sensitivity analysis. In Model 1 unadjusted for variables, the HRs (95% CI) for the middle RAR group and high RAR group were 2.02 (1.16–3.51), and 4.24 (2.56–7.04), respectively, compared with the low RAR group ($P < 0.05$). Even after adjusting for age, gender, hemoglobin, MCHC, blood urea nitrogen, chloride, congestive heart failure, hypertension, atrial flutter/fibrillation, diabetes with complications, and renal disease, the risk of 1-year mortality was significantly increased in the high RAR group compared with the low RAR group (HR = 2.21, 95% CI: 1.23–3.95, $P = 0.008$).

The Kaplan-Meier survival curves also revealed that patients in the high RAR group had a significantly higher mortality rate (log-rank $P < 0.001$, Fig 1).

### Dose-response relationship between RAR and mortality

The RCS regression model revealed a continuous linear relationship between RAR and 1-year all-cause mortality after adjusting for all covariates in Model 2, with higher RAR linked to an elevated mortality risk ($P$ for non-linearity = 0.970, Fig 2).

**Table 1. Baseline characteristics of participants.**

| Characteristics | Total (n = 760) | RAR | | | P value |
|---|---|---|---|---|---|
| | | < 3.5 (n = 254) | 3.5–4.0 (n = 253) | > 4.0 (n = 253) | |
| **Demographics characteristics** | | | | | |
| Age, years | 84.0 (77.0, 88.0) | 84.0 (77.0, 88.0) | 84.0 (77.0, 89.0) | 83.0 (76.0, 88.0) | 0.466 |
| Male, n (%) | 402 (52.9) | 128 (50.4) | 140 (55.3) | 134 (53.0) | 0.537 |
| BMI, kg/m$^2$ | 28.2±6.4 | 28.2±5.5 | 28.3±6.6 | 28.2±7.1 | 0.982 |
| **Laboratory parameters** | | | | | |
| RAR | 3.7 (3.3, 4.3) | 3.1 (3.0, 3.3) | 3.7 (3.6, 3.8) | 4.6 (4.3, 5.0) | < 0.001 |
| RDW, % | 14.6 (13.6, 15.9) | 13.4 (13.0, 13.9) | 14.7 (14.0, 15.5) | 16.4 (15.2, 17.8) | < 0.001 |
| Albumin, g/dL | 4.0 (3.7, 4.3) | 4.3 (4.1, 4.5) | 4.0 (3.8, 4.2) | 3.5 (3.3, 3.8) | < 0.001 |
| WBC, 10$^9$/L | 7.0 (5.5, 8.6) | 6.8 (5.7, 8.4) | 7.0 (5.6, 8.4) | 7.2 (5.1, 9.3) | 0.617 |
| RBC, 10$^{12}$/L | 3.7±0.6 | 3.8±0.5 | 3.7±0.6 | 3.5±0.7 | < 0.001 |
| Platelet, 10$^9$/L | 183.0 (144.0, 226.2) | 175.0 (141.2, 212.0) | 184.0 (147.0, 226.0) | 192.0 (150.0, 241.0) | 0.006 |
| Hematocrit, % | 34.1±5.3 | 35.7±4.6 | 34.5±5.0 | 32.1±5.5 | < 0.001 |
| Hemoglobin, g/dL | 11.0±1.9 | 11.8±1.6 | 11.1±1.7 | 10.1±1.9 | < 0.001 |
| MCV, fL | 92.6±6.5 | 93.5±4.9 | 92.7±6.2 | 91.7±7.9 | 0.009 |
| MCH, pg | 29.8±2.6 | 30.7±1.8 | 29.8±2.4 | 28.8±3.2 | < 0.001 |
| MCHC, g/dL | 32.2±1.5 | 32.9±1.2 | 32.2±1.3 | 31.4±1.6 | < 0.001 |
| Anion gap, mEq/L | 14.3±2.9 | 13.8±3.1 | 14.3±2.8 | 14.7±2.9 | < 0.001 |
| Bicarbonate, mEq/L | 25.7±3.8 | 25.5±3.6 | 25.8±3.6 | 25.7±4.2 | 0.719 |
| Blood urea nitrogen, mg/dL | 24.0 (18.0, 32.0) | 21.0 (16.0, 29.0) | 25.0 (18.0, 32.0) | 27.0 (20.0, 42.0) | < 0.001 |
| Chloride, mEq/L | 102.0 (99.0, 105.0) | 103.0 (100.0, 105.0) | 102.0 (100.0, 105.0) | 101.0 (98.0, 104.0) | 0.003 |
| Creatinine, mg/dL | 1.1 (0.9, 1.4) | 1.0 (0.8, 1.2) | 1.1 (0.9, 1.4) | 1.2 (0.9, 1.7) | < 0.001 |
| Glucose, mg/dL | 115.0 (99.0, 145.0) | 117.0 (101.0, 140.8) | 115.0 (99.2, 143.0) | 113.0 (97.2, 160.0) | 0.897 |
| Sodium, mEq/L | 140.0 (137.0, 141.2) | 140.0 (138.0, 142.0) | 140.0 (138.0, 142.0) | 139.0 (136.0, 141.0) | 0.010 |
| Potassium, mEq/L | 4.2 (3.9, 4.5) | 4.2 (3.9, 4.6) | 4.2 (3.9, 4.5) | 4.2 (3.9, 4.5) | 0.519 |
| **Comorbidities, n (%)** | | | | | |
| Myocardial infarct | 157 (20.7) | 47 (18.5) | 51 (20.2) | 59 (23.3) | 0.396 |
| Congestive heart failure | 540 (71.1) | 135 (53.1) | 181 (71.5) | 224 (88.5) | < 0.001 |
| Hypertension | 266 (35.0) | 111 (43.7) | 91 (36.0) | 64 (25.3) | < 0.001 |
| Atrial fibrillation/flutter | 356 (46.8) | 100 (39.4) | 119 (47.0) | 137 (54.2) | 0.004 |
| Peripheral vascular disease | 154 (20.3) | 57 (22.4) | 46 (18.2) | 51 (20.2) | 0.490 |
| Cerebrovascular disease | 103 (13.6) | 30 (11.8) | 42 (16.6) | 31 (12.3) | 0.220 |
| Pulmonary disorder | 133 (17.5) | 27 (10.6) | 45 (17.8) | 61 (24.1) | < 0.001 |
| Rheumatic disease | 62 (8.2) | 10 (3.9) | 26 (10.3) | 26 (10.3) | 0.011 |
| Liver disease | 55 (7.2) | 10 (3.9) | 14 (5.5) | 31 (12.3) | 0.001 |
| Diabetes | | | | | |
| without complications | 157 (20.7) | 51 (20.1) | 62 (24.5) | 44 (17.4) | 0.136 |
| with complications | 133 (17.5) | 41 (16.1) | 33 (13.0) | 59 (23.3) | 0.008 |
| Renal disease | 288 (37.9) | 73 (28.7) | 86 (34.0) | 129 (51.0) | < 0.001 |
| Charlson comorbidity index | 7.2±2.0 | 6.6±2.0 | 7.2±1.9 | 7.9±2.0 | < 0.001 |
| **Complication, n (%)** | | | | | |
| Acute kidney injury | 142 (18.7) | 32 (12.6) | 35 (13.8) | 75 (29.6) | < 0.001 |
| Blood transfusion | 150 (19.7) | 32 (12.6) | 41 (16.2) | 77 (30.4) | < 0.001 |
| Cardiac arrest | 27 (3.6) | 8 (3.1) | 9 (3.6) | 10 (4.0) | 0.888 |
| Cardiogenic shock | 27 (3.6) | 3 (1.2) | 10 (4.0) | 14 (5.5) | 0.028 |
| TIA/stroke | 37 (4.9) | 15 (5.9) | 11 (4.3) | 11 (4.3) | 0.642 |
| CRRT | 17 (2.2) | 2 (0.8) | 6 (2.4) | 9 (3.6) | 0.107 |

*(Continued)*

**Table 1.** (Continued)

| Characteristics | Total (n = 760) | RAR | | | P value |
|---|---|---|---|---|---|
| | | < 3.5 (n = 254) | 3.5–4.0 (n = 253) | > 4.0 (n = 253) | |
| Outcomes | | | | | |
| 30-day mortality, n (%) | 29 (3.8) | 5 (2.0) | 8 (3.2) | 16 (6.3) | 0.030 |
| 1-year mortality, n (%) | 127 (16.7) | 19 (7.5) | 37 (14.6) | 71 (28.1) | < 0.001 |
| Length of hospital stay, days | 5.0 (3.0, 8.2) | 4.0 (2.0, 7.0) | 5.0 (3.0, 8.0) | 7.0 (4.0, 12.0) | < 0.001 |
| ICU, n (%) | 481 (63.3) | 133 (52.4) | 164 (64.8) | 184 (72.7) | < 0.001 |

Notes: Data are presented as the mean ± SD, median (IQR), and n (%).

Abbreviations: BMI, body mass index; RAR, red blood cell distribution width-to-albumin ratio; RDW, red blood cell distribution width; WBC, white blood cells; RBC, red blood cells; MCV, mean corpuscular volume; MCH, mean corpuscular hemoglobin; MCHC, mean corpuscular hemoglobin concentration; TIA, transient ischemic attack; CRRT, continuous renal replacement therapy; ICU, intensive care unit.

## Subgroup analysis

We performed subgroup analysis to investigate the relationship between RAR and the primary outcome. Fig 3 depicts the results of subgroup analysis. The related common comorbidities were used as stratification variables, but no significant interaction was found.

## Association between RAR and length of hospital stay

Additionally, linear regression was utilized to assess the association between RAR and length of hospital stay. Even in the adjusted model, the RAR was related to the hospital stay length in patients who survived (β = 2.43, $P < 0.001$, Table 3). A sensitivity analysis using RAR as a categorical variable yielded similar results.

## Discussion

This study for the first time found that RAR was an independent risk factor for the increase of the 1-year mortality in patients undergoing TAVR. Additionally, a high level of RAR also lengthened the hospital stay. The subgroup analysis revealed a steady association between RAR and all-cause mortality.

Even though TAVR can significantly reduce mortality and improve prognosis, some patients have died or have to be readmitted in the first year. In the present study, 16.7% of the patients died within 1-year after TAVR, similar to previous studies [4]. Identifying patients

**Table 2.** HR (95% CI) for all-cause mortality across groups of RAR.

| Variable | Model 1 | | Model 2 | |
|---|---|---|---|---|
| | HR (95% CI) | P value | HR (95% CI) | P value |
| RAR | 1.78 (1.53 ∼ 2.07) | < 0.001 | 1.46 (1.22 ∼ 1.75) | < 0.001 |
| RAR Tertile | | | | |
| < 3.5 | Ref. | | Ref. | |
| 3.5–4.0 | 2.02 (1.16 ∼ 3.51) | 0.013 | 1.42 (0.80 ∼ 2.51) | 0.236 |
| > 4.0 | 4.24 (2.56 ∼ 7.04) | < 0.001 | 2.21 (1.23 ∼ 3.95) | 0.008 |
| P for trend | | < 0.001 | | 0.004 |

Notes: Cox proportional hazards regression models were used to calculate hazard ratios (HR) with 95% confidence intervals (CI); Model 1 covariates were adjusted for nothing; Model 2 covariates were adjusted for age, gender, hemoglobin, mean corpuscular hemoglobin concentration, blood urea nitrogen, chloride, congestive heart failure, hypertension, atrial flutter/fibrillation, diabetes with complications, and renal disease. RAR, red blood cell distribution width-to-albumin ratio.

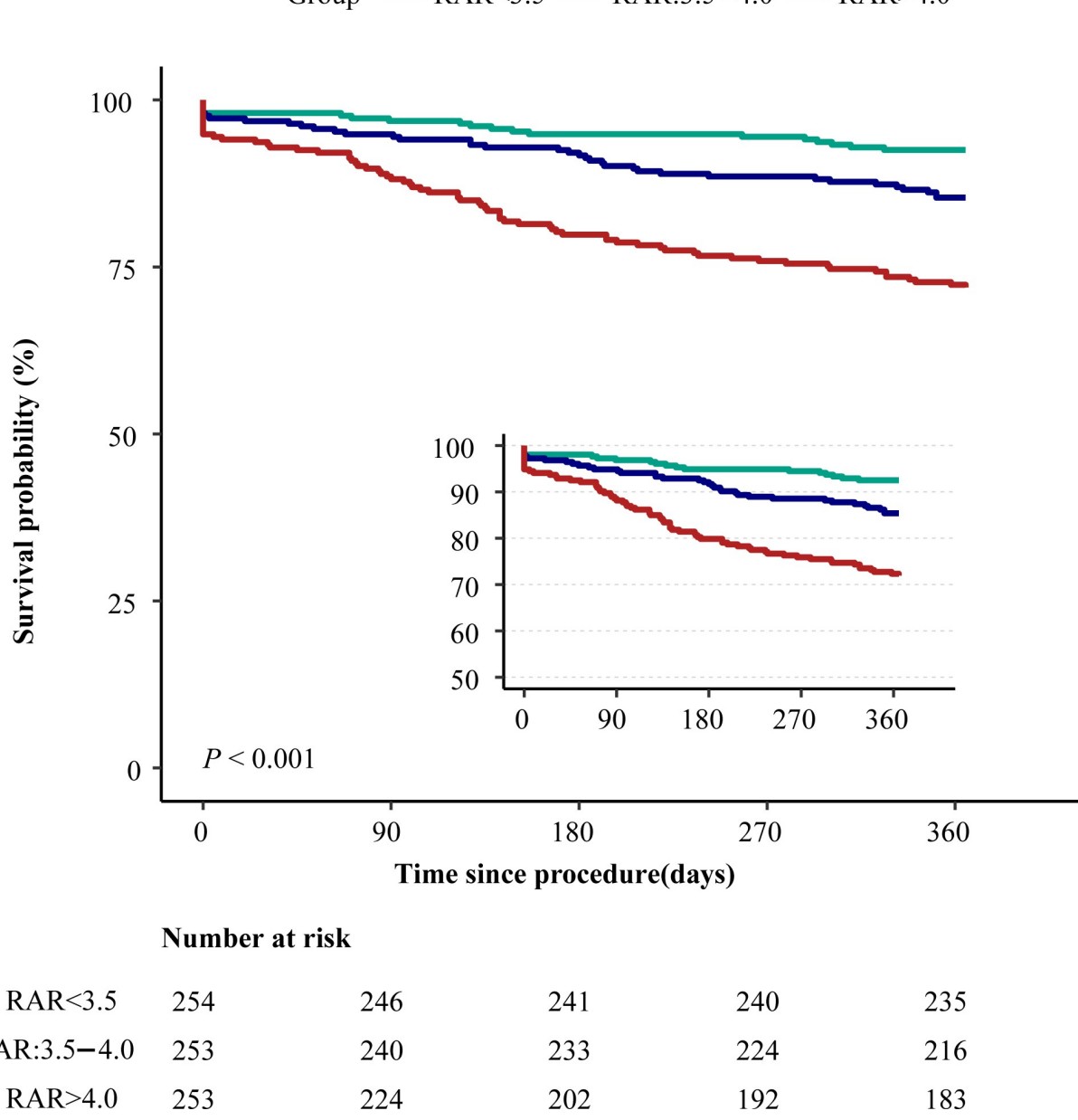

**Fig 1. Kaplan-Meier survival curves for all-cause mortality.** RAR, red blood cell distribution width-to-albumin ratio.

with a poor prognosis after successful TAVR can help decision-makers improve the preoperative conditions of patients, facilitate timely individualized intervention, and close postoperative follow-up to reduce adverse events.

Frailty is an aging syndrome resulting from a decline in the physiological reserve function of several organs [24]. For an aging population, frailty is becoming a more serious health problem in patients with cardiovascular disease [25]. Elderly patients undergoing TAVR frequently exhibit frailty, which is associated with mortality and unplanned hospital readmission [6]. According to a meta-analysis, frailty is associated with a worse early and

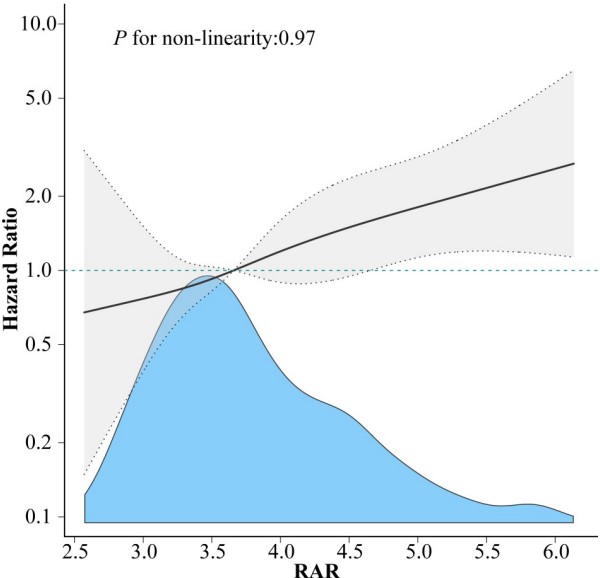

**Fig 2. Dose-response relationship between RAR and the risk of all-cause mortality.** Adjusted for all covariates in Model 2. The solid and dashed lines represent the estimated values and 95% confidence intervals. RAR, red blood cell distribution width-to-albumin ratio.

late prognosis in TAVR patients. Frailty individuals have a higher risk of late mortality than non-frailty patients [7].

It is essential to evaluate frailty in patients undergoing TAVR objectively. There is a strong relationship between inflammation and degenerative aortic valve disease [8–10]. Also, frailty results from the interaction of multiple factors, such as chronic inflammation, malnutrition, and various diseases. According to recent research, RDW is a nonspecific biomarker of systemic inflammation [11]. Increased RDW occurs due to inflammation that affects bone marrow function and inhibits RBC maturation. Furthermore, levels of RDW are increased in frailty patients, and a higher RDW is an indicator of frailty [12, 26]. According to previous studies, elevated RDW is a robust independent predictor of mortality after TAVR [27]. High baseline RDW was related to more adverse events and death after TAVR and could improve preoperative risk assessment in potential TAVR candidates when combined with the classical STS score [28].

Serum albumin, an indicator of the severity of malnutrition and chronic inflammation, is often used as a part of the frailty criteria [29]. Albumin is a strong predictor of death in many conditions, and hypoalbuminemia is common in elderly individuals [30]. A meta-analysis provided the most substantial evidence supporting the notion that serum albumin is a valuable prognostic marker in patients receiving TAVR, with low levels indicating a poor prognosis [31].

The RAR contains RDW and albumin measurements to provide inflammatory and nutritional information, reflecting key components of frailty. High RAR is associated with increased RDW and decreased albumin. We hypothesize that the mechanism by which RAR predicts mortality and readmission is due to the intricate interplay between nutrition, erythropoiesis, oxidative stress, and inflammation. The current study discovered that a higher RAR was related to a higher risk of 1-year mortality in patients undergoing TAVR. Considering early mortality is perhaps a bit more immune to other external factors affecting the bigger picture, it is possible that if a patient dies on the day of TAVR or the day after, this more likely reflects a

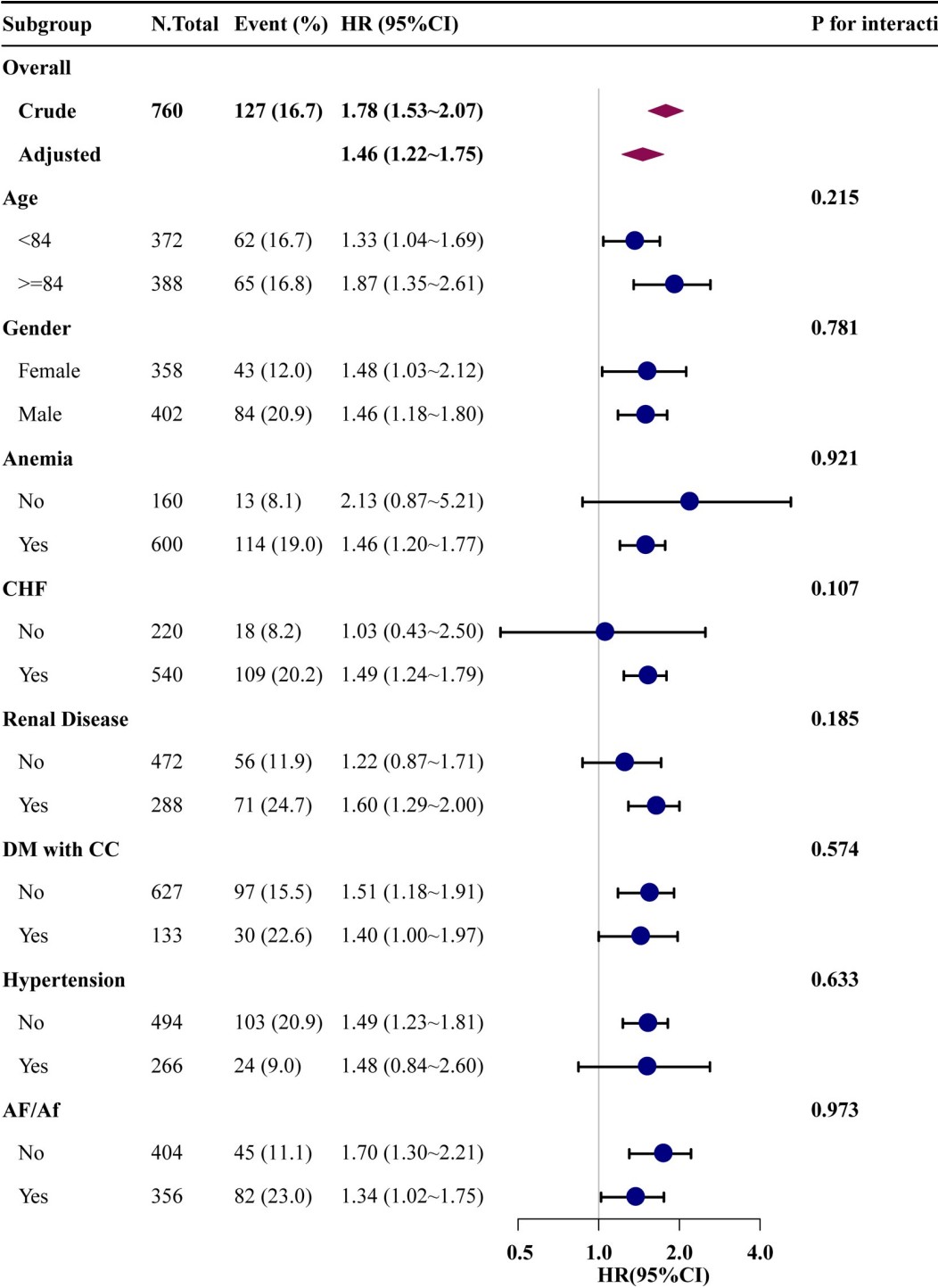

| Subgroup | N.Total | Event (%) | HR (95%CI) | P for interaction |
|---|---|---|---|---|
| **Overall** | | | | |
| Crude | 760 | 127 (16.7) | 1.78 (1.53~2.07) | |
| Adjusted | | | 1.46 (1.22~1.75) | |
| **Age** | | | | 0.215 |
| <84 | 372 | 62 (16.7) | 1.33 (1.04~1.69) | |
| >=84 | 388 | 65 (16.8) | 1.87 (1.35~2.61) | |
| **Gender** | | | | 0.781 |
| Female | 358 | 43 (12.0) | 1.48 (1.03~2.12) | |
| Male | 402 | 84 (20.9) | 1.46 (1.18~1.80) | |
| **Anemia** | | | | 0.921 |
| No | 160 | 13 (8.1) | 2.13 (0.87~5.21) | |
| Yes | 600 | 114 (19.0) | 1.46 (1.20~1.77) | |
| **CHF** | | | | 0.107 |
| No | 220 | 18 (8.2) | 1.03 (0.43~2.50) | |
| Yes | 540 | 109 (20.2) | 1.49 (1.24~1.79) | |
| **Renal Disease** | | | | 0.185 |
| No | 472 | 56 (11.9) | 1.22 (0.87~1.71) | |
| Yes | 288 | 71 (24.7) | 1.60 (1.29~2.00) | |
| **DM with CC** | | | | 0.574 |
| No | 627 | 97 (15.5) | 1.51 (1.18~1.91) | |
| Yes | 133 | 30 (22.6) | 1.40 (1.00~1.97) | |
| **Hypertension** | | | | 0.633 |
| No | 494 | 103 (20.9) | 1.49 (1.23~1.81) | |
| Yes | 266 | 24 (9.0) | 1.48 (0.84~2.60) | |
| **AF/Af** | | | | 0.973 |
| No | 404 | 45 (11.1) | 1.70 (1.30~2.21) | |
| Yes | 356 | 82 (23.0) | 1.34 (1.02~1.75) | |

HR(95%CI)

**Fig 3. Results from subgroup analysis showing the relationship between RAR and all-cause mortality.** Each stratification adjusted for all covariates in Model 2. CHF, congestive heart failure; DM with CC, diabetes with complications; AF/Af, atrial flutter/fibrillation.

technical complication or an acute deterioration. To address this, we performed a sensitivity analysis excluding patients who died within 30 days; in this sensitivity analysis, the association between RAR and death remained stable. The 1-year outcomes are also linked to the threshold

**Table 3. β (95% CI) for length of hospital stay across groups of RAR.**

| Variable | Model 1 | | Model 2 | |
|---|---|---|---|---|
| | β (95% CI) | *P* value | β (95% CI) | *P* value |
| RAR | 2.87 (2.20 ∼ 3.54) | < 0.001 | 2.43 (1.68 ∼ 3.19) | < 0.001 |
| RAR Tertile | | | | |
| < 3.5 | Ref. | | Ref. | |
| 3.5–4.0 | 1.53 (0.11 ∼ 2.94) | 0.034 | 1.21 (-0.21 ∼ 2.62) | 0.096 |
| > 4.0 | 4.85 (3.44 ∼ 6.26) | < 0.001 | 3.76 (2.15 ∼ 5.37) | < 0.001 |
| *P* for trend | | < 0.001 | | < 0.001 |

**Notes:** Multivariable linear regression models were used to calculate β with 95% confidence intervals (CI); Model 1 covariates were adjusted for nothing; Model 2 covariates were adjusted for age, gender, hemoglobin, mean corpuscular hemoglobin concentration, blood urea nitrogen, chloride, congestive heart failure, hypertension, atrial flutter/fibrillation, diabetes with complications, and renal disease. RAR, red blood cell distribution width-to-albumin ratio.

for offering this treatment. These include the national wealth, funding models, physician incentives, the timing of referral and intervention, etc. Future studies therefore are needed to further elucidate such relationship. In addition, we found that patients with elevated RAR had a significantly longer hospital stay. However, the high-RAR group had a twice-higher rate of acute kidney injury and blood transfusion after TAVR. Moreover, they had a four to five times higher rate of cardiogenic shock. These complications might be the main cause of the longer hospital stay in the high-RAR group. From the clinical viewpoint, these complications are related to patients' anatomy, ejection fraction, and technical aspects. Therefore, future studies are required for further clinical validation. Although there are other scoring methods for predicting the prognosis of patients undergoing TAVR, RAR remains helpful since it can be easily and quickly obtained from the admission laboratory without additional expense and complex calculations. Therefore, RAR could be a simple, objective, but relatively reliable indicator for preoperative risk stratification in clinical decision-making. RAR can assist in identifying which subjects will gain the most from TAVR. In addition, it can also be used to identify patients who require close monitoring to live longer. In other words, not only can RAR help with appropriate patient selection, but the focus should be on better follow-up to improve clinical outcomes.

The underlying mechanisms that relate RAR to poor endpoints in the TAVR population remain unknown. RDW appears modifiable through exercise training, which has been demonstrated to reduce RDW in individuals with coronary artery disease, and iron therapy, which has been shown to reduce RDW in hemodialysis patients [32, 33]. Hypoproteinemia can be corrected by albumin infusion and improving the patient's diet. However, more study is needed to determine the underlying pathophysiological mechanisms and whether aggressive therapies can improve the prognosis of individuals with high RAR.

Our study has several limitations. Firstly, there may be selection bias since this is a single-center retrospective analysis. Nonetheless, validation in larger multicentric cohorts is needed to further sustain our findings. Secondly, some variables, such as echocardiographic data, are not included owing to missing data in the database, which may impact the result. Since pre-validated frailty scores are not recorded in the MIMIC-IV database, we were not able to compare the effectiveness of RAR with that of other conventional parameters such as the clinical frailty scale. Thirdly, we only adopted baseline RAR before TAVR, and dynamic changes may have more predictive value. However, because the purpose of the study was to assist clinicians with preoperative assessments, postoperative changes were less critical in the decision-making

process. Despite these limitations, our study demonstrates for the first time that the RAR can provide risk stratification for patients receiving TAVR.

## Conclusions

The RAR was independently associated with all-cause mortality in patients undergoing TAVR. The RAR is a simple, objective tool to assess frailty that could help with the risk stratification of TAVR patients. Our findings need to be further validated by large prospective studies.

## Supporting information

**S1 Table. Association of RAR with all-cause mortality after excluding patients who died within 30 days. Notes:** Cox proportional hazards regression models were used to calculate hazard ratios (HR) with 95% confidence intervals (CI); Model 1 covariates were adjusted for nothing; Model 2 covariates were adjusted for age, gender, hemoglobin, mean corpuscular hemoglobin concentration, blood urea nitrogen, chloride, congestive heart failure, hypertension, atrial flutter/fibrillation, diabetes with complications, and renal disease. RAR, red blood cell distribution width-to-albumin ratio.
(DOCX)

## Acknowledgments

We would like to thank the administrators of the MIMIC-IV database for data support.

## Author Contributions

**Conceptualization:** Guoqiang Gu.

**Data curation:** Limin Meng, Hua Yang, Chao Chang.

**Formal analysis:** Limin Meng, Hua Yang, Chao Chang, Lijun Liu.

**Investigation:** Limin Meng, Hua Yang.

**Methodology:** Shuanli Xin, Guoqiang Gu.

**Supervision:** Guoqiang Gu.

**Writing – original draft:** Limin Meng, Hua Yang, Lijun Liu.

**Writing – review & editing:** Shuanli Xin, Guoqiang Gu.

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
