## [Decision Letter · Decision Letter 0]

20 Mar 2023

PONE-D-23-03232Association of red blood cell distribution width-to-albumin ratio with mortality in patients undergoing transcatheter aortic valve replacementPLOS ONE

Dear Dr. Gu,

Thank you for submitting your manuscript to PLOS ONE. After careful consideration, we feel that it has merit but does not fully meet PLOS ONE’s publication criteria as it currently stands. Therefore, we invite you to submit a revised version of the manuscript that addresses the points raised during the review process.

We look forward to receiving your revised manuscript.

Kind regards,

Satoshi Higuchi

Academic Editor

PLOS ONE

Journal Requirements:

Reviewers' comments:

Reviewer's Responses to Questions

**Comments to the Author**

1. Is the manuscript technically sound, and do the data support the conclusions?

Reviewer #1: Yes

Reviewer #2: Yes

2. Has the statistical analysis been performed appropriately and rigorously? 

Reviewer #1: Yes

Reviewer #2: Yes

3. Have the authors made all data underlying the findings in their manuscript fully available?

Reviewer #1: Yes

Reviewer #2: No

4. Is the manuscript presented in an intelligible fashion and written in standard English?

Reviewer #1: Yes

Reviewer #2: Yes

5. Review Comments to the Author

Reviewer #1: This is a well written manuscript that shows relationship between RAR and mortality after TAVR. Although this manuscript is well written, several points should be added to improve the paper.

First, I totally agree with the author’s idea that RAR, as the marker for frailty, is associated with 1-year mortality after TAVR. However, when it comes to secondary outcome, length of hospitalization, it is unclear to me. According to Table 1, high RAR group had twice higher rate of acute kidney injury and blood transfusion after TAVR. In addition, they had four to five times higher rate of cardiogenic shock. These complications should be the main cause of longer hospital stay in high RAR group. Do the authors consider these complications are related to high RAR or frailty of the patients? From the clinical viewpoint, these complications are related to patients’ anatomy, EF, and technical aspects.

Second, although RAR seems to be a good indicator for poor outcomes after TAVR, we usually evaluate patients physically before the procedure to assess the possible risks for the procedure. If TAVR is an emergent procedure and we do not have enough time to assess patients’ clinical frailty, RAR seems to be a good alternative way to assess it. However, TAVR is usually performed as non-emergent procedure and we cardiologist should assess patients carefully before the procedure. Considering this, what is the clinical implication for assessing RAR in addition to assessing routine parameters such as clinical frailty scale? Did the authors compare effectiveness of RAR with that of other conventional parameters such as clinical frailty scale?

Third, high RAR group showed very high 1-year mortality (28%) and high complication rates (30% acute kidney disease, 30% transfusion, etc.). Do the authors consider to avoid these patients to offer TAVR?

Reviewer #2: A topical and well written paper. It comes on the back of studying the predictive value of RDW and RAR in relation to various health conditions and/or interventions in elderly and frail patients. The paper shows elegantly that RAR is an independent predictor of 1-year survival after TAVR. A few questions, in no particular order.

1. Premise. Would it be useful to also report some data on 30-day mortality? The striking observation in the bigger picture is that 16.7% of patients have died 1 year after this expensive intervention. They are clearly in serious decline if not even this kind of expensive treatment can prolong life significantly. The 1-year outcomes are also linked to the threshold for offering this treatment: national wealth, funding models, physician incentives, timing of referral and intervention etc. In that sense echo data would have been interesting but it is understandable why they cannot be reported. Early mortality is perhaps a bit more immune to other external factors affecting the bigger picture. Very early mortality, procedure-related, may also allow useful exclusions when focusing on RAR. If a patient dies on the day of TAVR or the day after this more likely reflects a technical complication or an acute deterioration that is less likely related to RAR. Perhaps some of these observations can find their way into the analysis, or at least into the discussion.

2. Validation. The authors suggest at the end that the model will be validated in their hospital. I suspect that the patient populations in the 2 hospitals are quite different and the biases suggested above are at play. Validation is more powerful if done in multicentric fashion.

3. Statistics. I am not qualified to comment, an expert review is needed for that. The paper may gain in clarity and decrease in length by choosing 1 or 2 models instead of 3. It would be reasonable to opt for the most complex or the most appropriate model from the outset and explain this choice. A shorter manuscript in this respect may create the opportunity to examine early outcomes (if available) and offer overall a better description of the TAVR journey.

In summary, a good paper that shows a clear relationship between RAR and TAVR survival at 1 year. This in itself is useful but we have to note that the field is changing rapidly in terms of indications, availability etc. This makes generalisations more difficult but clearly this kind of indicator will continue to play a part. Its simplicity is also appealing. One thing that should be noted is that it usefulness can also show in patient selection (I am not sure I have seen this mentioned in the paper). In other words it may help turn some patients down, it cannot be assumed that all comers will get this treatment and the focus should be only on better follow up to improve outcomes. The paper may become more visible by adopting a few small changes.

6. PLOS authors have the option to publish the peer review history of their article (what does this mean?). If published, this will include your full peer review and any attached files.

Reviewer #1: No

Reviewer #2: No

---

## [Author Response · Author response to Decision Letter 0]

4 Apr 2023

Re: Manuscript Number: PONE-D-23-03232

Title: Association of red blood cell distribution width-to-albumin ratio with mortality in patients undergoing transcatheter aortic valve replacement

Dear Editor and Reviewers， 

On behalf of my co-authors, we greatly appreciate the careful review and constructive comments from you and the reviewers. We believe that by implementing the suggested changes, we now have a stronger manuscript entitled “Association of red blood cell distribution width-to-albumin ratio with mortality in patients undergoing transcatheter aortic valve replacement” (Manuscript Number: PONE-D-23-03232) for submission to PLOS ONE. 

We have provided point-by-point responses to the reviewer comments and have revised the manuscript accordingly. And we hope the revised manuscript will be acceptable to you. All the changes are indicated in the revised manuscript using track changes. The line and page numbers indicated in our response refer to the revised manuscript.

There are no conflicts of interest regarding this work. All authors have read the revised manuscript and approved its submission to PLOS ONE. Once again, we would like to express our great appreciation to you and reviewers for comments. Looking forward to hearing from you.

Thank you and best regards. 

Yours Sincerely, 

Guoqiang Gu 

Email: guguoqiang2022@163.com. 

Department of Cardiology, The Second Hospital of Hebei Medical University, Shijiazhuang, Hebei, China

Below are our specific responses to the editor’s comments.

Responses to the Editor:

Response 1: We followed the editor’s suggestions and made the corresponding changes to the revised version. 

2.Please provide additional details regarding participant consent. In the ethics statement in the Methods and online submission information, please ensure that you have specified (1) whether consent was informed and (2) what type you obtained (for instance, written or verbal, and if verbal, how it was documented and witnessed). If your study included minors, state whether you obtained consent from parents or guardians. If the need for consent was waived by the ethics committee, please include this information.

Response 2: Thank you very much for your proposal. All patient-related information in the database is anonymous, so informed consent was waived. We have added the full name of the ethics committee that waived the need for informed consent to the ethics statement of the Methods and online submission information: “The study was approved and granted a waiver of informed consent by the institutional review boards of the Massachusetts Institute of Technology (MIT) and Beth Israel Deaconess Medical Center (BIDMC).” (Page 4, lines 76-78) 

3.Please include your full ethics statement in the ‘Methods’ section of your manuscript file. In your statement, please include the full name of the IRB or ethics committee who approved or waived your study, as well as whether or not you obtained informed written or verbal consent. If consent was waived for your study, please include this information in your statement as well. 

Response 3: We appreciate your proposition. We have included the full name of the IRB that approved our study and waived the need for informed consent in the ‘Methods’ section of my manuscript and my ethics statement as well: “The study was approved and granted a waiver of informed consent by the institutional review boards of the Massachusetts Institute of Technology (MIT) and Beth Israel Deaconess Medical Center (BIDMC).” (Page 4, lines 76-78)

Below are our specific responses to the reviewers’ comments.

Responses to Reviewer #1: 

This is a well written manuscript that shows relationship between RAR and mortality after TAVR. Although this manuscript is well written, several points should be added to improve the paper.

Comment 1: 

First, I totally agree with the author’s idea that RAR, as the marker for frailty, is associated with 1-year mortality after TAVR. However, when it comes to secondary outcome, length of hospitalization, it is unclear to me. According to Table 1, high RAR group had twice higher rate of acute kidney injury and blood transfusion after TAVR. In addition, they had four to five times higher rate of cardiogenic shock. These complications should be the main cause of longer hospital stay in high RAR group. Do the authors consider these complications are related to high RAR or frailty of the patients? From the clinical viewpoint, these complications are related to patients’ anatomy, EF, and technical aspects.

Response 1: 

Thank you very much for your careful review and constructive suggestions with regard to our manuscript. This is an especially important issue. We agree with the reviewer’s comment that these complications should be the main cause of the longer hospital stay in the high RAR group. We have added the content to the discussion section of the revised manuscript. The details are as follows: 

In addition, we found that patients with elevated RAR had a significantly longer hospital stay. However, the high-RAR group had a twice-higher rate of acute kidney injury and blood transfusion after TAVR. Moreover, they had a four to five times higher rate of cardiogenic shock. These complications might be the main cause of the longer hospital stay in the high-RAR group. From the clinical viewpoint, these complications are related to patients’ anatomy, ejection fraction, and technical aspects. Therefore, future studies are required for further clinical validation. (Page 14, lines 262-266, and Page 15, lines 267-269)

Second, we further performed analysis to explore the association between RAR and these complications using the multivariable logistic regression model. The results were presented as odds ratios (OR) with 95% confidence intervals (CI). The results showed that these complications are related to RAR levels (Table S1). However, residual confounding may have occurred since we did not collect data on variables known to be associated with complications, such as patients’ anatomy, EF, and technical aspects. Therefore, this result will require further validation in the future.

Table S1. Univariable and multivariable analyses for the relationship between the complications and RAR.

Variables Univariable model Multivariable model

 OR (95% CI) P value OR (95% CI) P value

Acute kidney injury 

RAR 1.66 (1.36 ∼2.04) <0.001 1.28 (0.99 ∼1.65) 0.055

Blood transfusion 

RAR 1.79 (1.46 ∼2.20) <0.001 1.47 (1.16∼1.88) 0.002

Cardiogenic shock 

RAR 1.89 (1.36 ∼2.61) <0.001 1.82 (1.21∼2.75) 0.004

Notes: Multivariable model covariates were adjusted for age, gender, hemoglobin, mean corpuscular hemoglobin concentration, blood urea nitrogen, chloride, congestive heart failure, hypertension, atrial flutter/fibrillation, diabetes with complications, and renal disease. 

Comment 2: 

Second, although RAR seems to be a good indicator for poor outcomes after TAVR, we usually evaluate patients physically before the procedure to assess the possible risks for the procedure. If TAVR is an emergent procedure and we do not have enough time to assess patients’ clinical frailty, RAR seems to be a good alternative way to assess it. However, TAVR is usually performed as non-emergent procedure and we cardiologist should assess patients carefully before the procedure. Considering this, what is the clinical implication for assessing RAR in addition to assessing routine parameters such as clinical frailty scale? Did the authors compare effectiveness of RAR with that of other conventional parameters such as clinical frailty scale?

Response 2:

Thank you for reviewing our manuscript and for the constructive comments, which greatly helped us to improve the manuscript. The Clinical Frailty Scale (CFS) is a simple scale and is widely used as a useful tool for risk evaluation before TAVR [1]. However, the CFS tool is semi-quantitative and subjective in nature, and therefore predisposed to inter-observer variability. Frailty assessment remains a clinical diagnosis prone to subjectivity. Specific evaluation of frailty should incorporate objective estimates that, ideally, are easy to obtain [2]. The RAR, a simple, objective, and readily available tool to assess frailty, might be incorporated into assessing patients being considered for TAVR. Future studies will explore whether the CFS combined with the RAR may be more effective in assessing frailty. 

We did not compare the effectiveness of RAR with that of other conventional parameters. We have acknowledged this in the limitations. The details are as follows:

 “Since pre-validated frailty scores are not recorded in the MIMIC-IV database, we were not able to compare the effectiveness of RAR with that of other conventional parameters such as the clinical frailty scale.” (Page 16, lines 290-292) 

References

1. Shimura T, Yamamoto M, Kano S, et al. Impact of the clinical frailty scale on outcomes after Transcatheter aortic valve replacement. Circulation. 2017;135: 2013-2024.

2. Baumgartner H, Falk V, Bax JJ, et al. 2017 ESC/EACTS Guidelines for the management of valvular heart disease. Eur Heart J. 2017;38(36):2739-2791.

Comment 3: 

Third, high RAR group showed very high 1-year mortality (28%) and high complication rates (30% acute kidney disease, 30% transfusion, etc.). Do the authors consider to avoid these patients to offer TAVR?

Response 3: 

Thank you for your careful review. Not only can RAR help with appropriate patient selection, but the focus should be on better follow-up to improve clinical outcomes. Assessing patients’ RAR prior to TAVR will positively affect their clinical care and improve postoperative outcomes. The RAR might be taken into account as part of the pre-operative risk assessment as a means to further improve patient management and optimize outcomes after TAVR. We have added this part to the “Discussion” part as follows: Not only can RAR help in patient selection, but the focus should be on better follow-up to improve clinical outcomes. (Page 15, lines 276-277)

Responses to Reviewer #2: 

A topical and well written paper. It comes on the back of studying the predictive value of RDW and RAR in relation to various health conditions and/or interventions in elderly and frail patients. The paper shows elegantly that RAR is an independent predictor of 1-year survival after TAVR. A few questions, in no particular order.

Comment 1: 

Premise. Would it be useful to also report some data on 30-day mortality? The striking observation in the bigger picture is that 16.7% of patients have died 1 year after this expensive intervention. They are clearly in serious decline if not even this kind of expensive treatment can prolong life significantly. The 1-year outcomes are also linked to the threshold for offering this treatment: national wealth, funding models, physician incentives, timing of referral and intervention etc. In that sense echo data would have been interesting but it is understandable why they cannot be reported. Early mortality is perhaps a bit more immune to other external factors affecting the bigger picture. Very early mortality, procedure-related, may also allow useful exclusions when focusing on RAR. If a patient dies on the day of TAVR or the day after this more likely reflects a technical complication or an acute deterioration that is less likely related to RAR. Perhaps some of these observations can find their way into the analysis, or at least into the discussion.

Response 1: 

Thank you very much for your careful review and constructive suggestions with regard to our manuscript. Meanwhile, I am also very grateful to you for the specific modification strategy. 

First, we agree with the reviewer, thus we have improved the discussion section:

The 1-year outcomes are also linked to the threshold for offering this treatment. These include the national wealth, funding models, physician incentives, the timing of referral and intervention, etc. Future studies therefore are needed to further elucidate such relationship. (Page 14, lines 259-262)

Second, we have supplemented some contents based on your advice. The number of 30-day deaths in this study was 29 (3.8%). At the same time, we have added this part to Table 1 in the revised manuscript. We agree with the reviewer that if a patient dies on the day of TAVR or the day after, this more likely reflects a technical complication or an acute deterioration that is less likely related to RAR. Therefore, we also did sensitivity analysis following your suggestions. In the sensitivity analysis excluding patients who died within 30 days postoperatively, RAR remained substantially associated with all-cause mortality (HR = 1.39, 95% CI: 1.08-1.79, P = 0.010) (S1 Table). We have included all these comments in the revised version of our manuscript. The details are as follows: 

In addition, we performed a sensitivity analysis, excluding deaths occurring within 30 days of enrolment to exclude deaths that could potentially be attributable to a technical complication or an acute deterioration. (Page 7, lines 127-129) 

After excluding patients who died within 30 days, the adjusted association remained significant (HR = 1.39, 95% CI: 1.08-1.79, P = 0.010) (S1 Table). (Page 10, lines 161-162)

Considering early mortality is perhaps a bit more immune to other external factors affecting the bigger picture, it is possible that if a patient dies on the day of TAVR or the day after, this more likely reflects a technical complication or an acute deterioration. To address this, we performed a sensitivity analysis excluding patients who died within 30 days; in this sensitivity analysis, the association between RAR and death remained stable. (Page 14, lines 254-259)

Table 1. Baseline characteristics of participants.

Characteristics Total

(n = 760) RAR P value

 <3.5 (n = 254) 3.5-4.0 (n = 253) >4.0 (n = 253) 

Outcomes 

30-day mortality, n (%) 29 (3.8) 5 (2.0) 8 (3.2) 16 (6.3) 0.030

S1 Table. Association of RAR with all-cause mortality after excluding patients who died within 30 days.

Variable Model 1 Model 2

 HR (95% CI) P value HR (95% CI) P value

RAR 1.70 (1.40∼2.07) <0.001 1.39 (1.08∼1.79) 0.010

RAR Tertile 

<3.5 Ref. Ref. 

3.5-4.0 2.17 (1.15∼4.10) 0.017 1.51 (0.78∼2.93) 0.226

>4.0 4.59 (2.55∼8.26) <0.001 2.34 (1.19∼4.57) 0.013

P for trend <0.001 0.009

Notes: Cox proportional hazards regression models were used to calculate hazard ratios (HR) with 95% confidence intervals (CI); Model 1 covariates were adjusted for nothing; Model 2 covariates were adjusted for age, gender, hemoglobin, mean corpuscular hemoglobin concentration, blood urea nitrogen, chloride, congestive heart failure, hypertension, atrial flutter/fibrillation, diabetes with complications, and renal disease.

RAR, red blood cell distribution width-to-albumin ratio.

Comment 2: 

Validation. The authors suggest at the end that the model will be validated in their hospital. I suspect that the patient populations in the 2 hospitals are quite different and the biases suggested above are at play. Validation is more powerful if done in multicentric fashion.

Response 2: 

We thank the reviewer for pointing out this issue. Based on your suggestion, we added it in limitations as “Nonetheless, validation in larger multicentric cohorts is needed to further sustain our findings.” (Page 15, lines 287-288)

Comment 3: 

Statistics. I am not qualified to comment, an expert review is needed for that. The paper may gain in clarity and decrease in length by choosing 1 or 2 models instead of 3. It would be reasonable to opt for the most complex or the most appropriate model from the outset and explain this choice. A shorter manuscript in this respect may create the opportunity to examine early outcomes (if available) and offer overall a better description of the TAVR journey.

Response 3: 

Thank you for your great suggestion. Based on your suggestion, we chose 2 models instead of 3 in the revised manuscript. We have made changes in the corresponding part. (Page 10, lines 172-173, and Page 12, lines 205-206)

Comment 4: 

In summary, a good paper that shows a clear relationship between RAR and TAVR survival at 1 year. This in itself is useful but we have to note that the field is changing rapidly in terms of indications, availability etc. This makes generalisations more difficult but clearly this kind of indicator will continue to play a part. Its simplicity is also appealing. One thing that should be noted is that its usefulness can also show in patient selection (I am not sure I have seen this mentioned in the paper). In other words, it may help turn some patients down, it cannot be assumed that all comers will get this treatment and the focus should be only on better follow up to improve outcomes. The paper may become more visible by adopting a few small changes.

Response 4: 

We would like to thank the reviewer for your thoughtful review of our manuscript. We fully agree with your suggestion and have added this part to the “Discussion” part as follows: Not only can RAR help in patient selection, but the focus should be on better follow-up to improve clinical outcomes. (Page 15, lines 276-277)

We appreciate your efforts in reviewing our manuscript, which have made our study clearer and more comprehensive. We believe that we have adequately responded to the editor’s and reviewers’ comments, and we hope that our paper is now acceptable for publication by PLOS ONE.

---

## [Decision Letter · Decision Letter 1]

11 Apr 2023

PONE-D-23-03232R1Association of red blood cell distribution width-to-albumin ratio with mortality in patients undergoing transcatheter aortic valve replacementPLOS ONE

Dear Dr. Gu,

Thank you for submitting your manuscript to PLOS ONE. After careful consideration, we feel that it has merit but does not fully meet PLOS ONE’s publication criteria as it currently stands. Therefore, we invite you to submit a revised version of the manuscript that addresses the points raised during the review process.

We look forward to receiving your revised manuscript.

Kind regards,

Satoshi Higuchi

Academic Editor

PLOS ONE

Journal Requirements:

Additional Editor Comments (if provided):Thank you for your response. Your manuscript has improved after constructive comments from the reviewers. I will not request further analysis. However, there are some simple mistakes. For example, you depicted an arrow in the bar graph in Figure 3 "anemia". Could you please review your manuscript carefully again?Thank you again for your nice manuscript.  

Reviewers' comments:

Reviewer's Responses to Questions

**Comments to the Author**

1. If the authors have adequately addressed your comments raised in a previous round of review and you feel that this manuscript is now acceptable for publication, you may indicate that here to bypass the “Comments to the Author” section, enter your conflict of interest statement in the “Confidential to Editor” section, and submit your "Accept" recommendation.

Reviewer #1: All comments have been addressed

Reviewer #2: All comments have been addressed

2. Is the manuscript technically sound, and do the data support the conclusions?

Reviewer #1: Yes

Reviewer #2: Yes

3. Has the statistical analysis been performed appropriately and rigorously? 

Reviewer #1: Yes

Reviewer #2: Yes

4. Have the authors made all data underlying the findings in their manuscript fully available?

Reviewer #1: Yes

Reviewer #2: Yes

5. Is the manuscript presented in an intelligible fashion and written in standard English?

Reviewer #1: Yes

Reviewer #2: Yes

6. Review Comments to the Author

Reviewer #1: Thank you for replying all of the comments. Since they replied all of my comments properly, I have no additional comments.

Reviewer #2: Thank you for being receptive. Thank you for being receptive. Thank you for being receptive. Thank you for being receptive.

7. PLOS authors have the option to publish the peer review history of their article (what does this mean?). If published, this will include your full peer review and any attached files.

Reviewer #1: No

Reviewer #2: No

---

## [Author Response · Author response to Decision Letter 1]

14 Apr 2023

Re: Manuscript Number: PONE-D-23-03232

Title: Association of red blood cell distribution width-to-albumin ratio with mortality in patients undergoing transcatheter aortic valve replacement

Dear Editor， 

On behalf of my co-authors, we thank you again very much for giving us the opportunity to revise our manuscript. We believe that by implementing the suggested changes, we now have a stronger manuscript entitled “Association of red blood cell distribution width-to-albumin ratio with mortality in patients undergoing transcatheter aortic valve replacement” (Manuscript Number: PONE-D-23-03232) for submission to PLOS ONE. 

We have revised the manuscript according to your comments. And we hope the revised manuscript will be acceptable to you. All the changes are indicated in the revised manuscript using track changes. The line and page numbers indicated in our response refer to the revised manuscript.

There are no conflicts of interest regarding this work. All authors have read the revised manuscript and approved its submission to PLOS ONE. Once again, we would like to express our great appreciation to you and reviewers for comments. Looking forward to hearing from you.

Thank you and best regards. 

Yours Sincerely, 

Guoqiang Gu 

Email: guguoqiang2022@163.com. 

Department of Cardiology, The Second Hospital of Hebei Medical University, Shijiazhuang, Hebei, China

Below are our specific responses to the editor’s comments.

Responses to the Editor:

1. Thank you for your response. Your manuscript has improved after constructive comments from the reviewers. I will not request further analysis. However, there are some simple mistakes. For example, you depicted an arrow in the bar graph in Figure 3 "anemia". Could you please review your manuscript carefully again?

Response 1: Thank you very much for your careful review. We have checked our manuscript carefully. We have remade and reuploaded the Figure 3 according to your suggestion. 

Below are our specific responses to the Journal Requirements.

Response 1: According to the requirements of the journal, we checked for the correctness and formatting of the references and updated them within the manuscript and the reference list. (Page 18, lines 326 and 327, lines 330 and 331) 

In addition, we did not cite papers that have been retracted.

We appreciate your efforts in reviewing our manuscript, which have made our study clearer and more comprehensive. We hope that our paper is now acceptable for publication by PLOS ONE.

---

## [Editor Report · Decision Letter 2]

7 May 2023

PONE-D-23-03232R2Association of red blood cell distribution width-to-albumin ratio with mortality in patients undergoing transcatheter aortic valve replacementPLOS ONE

Dear Dr. Gu,

Thank you for submitting your manuscript to PLOS ONE. Your manuscript is almost acceptable. However, the authors should express gratitude for the administrators of the database MIMIC IV in the Acknowledge section. 

We look forward to receiving your revised manuscript.

Kind regards,

Satoshi Higuchi

Academic Editor

PLOS ONE
---

## [Author Response · Author response to Decision Letter 2]

8 May 2023

Re: Manuscript Number: PONE-D-23-03232

Title: Association of red blood cell distribution width-to-albumin ratio with mortality in patients undergoing transcatheter aortic valve replacement

Dear Editor, 

Thank you again very much. We have revised the manuscript according to your suggestions. And we hope the revised manuscript will be acceptable to you. All the changes are indicated in the revised manuscript using track changes. The line and page numbers indicated in our response refer to the revised manuscript.

There are no conflicts of interest regarding this work. All authors have read the revised manuscript and approved its submission to PLOS ONE. Once again, we would like to express our great appreciation to you and the reviewers for comments. Looking forward to hearing from you.

Thank you and best regards. 

Yours Sincerely, 

Guoqiang Gu 

Email: guguoqiang2022@163.com. 

Department of Cardiology, The Second Hospital of Hebei Medical University, Shijiazhuang, Hebei, China

 

Below are our specific responses to the editor’s comments.

Responses to the Editor:

1. The authors should express gratitude for the administrators of the database MIMIC IV in the Acknowledge section. 

Response 1: Thank you very much for your careful review. We have supplemented this part in the revised manuscript. (Page 16, lines 306 and 307) 

Below are our specific responses to the Journal Requirements.

Response 1: According to the requirements of the journal, we checked for the correctness and formatting of the references. In addition, we did not cite papers that have been retracted.

We appreciate your efforts in reviewing our manuscript, which have made our study clearer and more comprehensive. We hope that our paper is now acceptable for publication by PLOS ONE.

---

## [Editor Report · Decision Letter 3]

19 May 2023

Association of red blood cell distribution width-to-albumin ratio with mortality in patients undergoing transcatheter aortic valve replacement

PONE-D-23-03232R3

Dear Dr. Gu,

We’re pleased to inform you that your manuscript has been judged scientifically suitable for publication and will be formally accepted for publication once it meets all outstanding technical requirements.

Kind regards,

Satoshi Higuchi

Academic Editor

PLOS ONE
---

## [Editor Report · Acceptance letter]

26 May 2023

PONE-D-23-03232R3 

Association of red blood cell distribution width-to-albumin ratio with mortality in patients undergoing transcatheter aortic valve replacement 

Dear Dr. Gu:

I'm pleased to inform you that your manuscript has been deemed suitable for publication in PLOS ONE. Congratulations! Your manuscript is now with our production department. 

Kind regards, 

on behalf of

Dr. Satoshi Higuchi 

Academic Editor

PLOS ONE